# The Influence of Chosen Plant Fillers in PHBV Composites on the Processing Conditions, Mechanical Properties and Quality of Molded Pieces

**DOI:** 10.3390/polym13223934

**Published:** 2021-11-14

**Authors:** Wiesław Frącz, Grzegorz Janowski, Robert Smusz, Marek Szumski

**Affiliations:** 1Department of Material Forming and Processing, Faculty of Mechanical Engineering and Aeronautics, Rzeszow University of Technology, 35-959 Rzeszów, Poland; gjan@prz.edu.pl; 2Department of Thermodynamics, Faculty of Mechanical Engineering and Aeronautics, Rzeszow University of Technology, 35-959 Rzeszów, Poland; robsmusz@prz.edu.pl; 3Department of Aerospace Engineering, Faculty of Mechanical Engineering and Aeronautics, Rzeszow University of Technology, 35-959 Rzeszów, Poland; szumarek@prz.edu.pl

**Keywords:** biopolymers, biocomposites, PHBV, plant fibers, extrusion process, injection molding process

## Abstract

This work is inspired by the current European policies that aim to reduce plastic waste. This is especially true of the packaging industry. The biocomposites developed in the work belong to the group of environmentally friendly plastics that can reduce the increasing costs of environmental fees in the future. Three types of short fibers (flax, hemp and wood) with a length of 1 mm each were selected as fillers (30% mass content in PHBV). The biocomposites were extruded and then processed by the injection molding process with the same technical parameters. The samples obtained in this way were tested for mechanical properties and quality of the molded pieces. A significant improvement of some mechanical properties of biocomposites containing hemp and flax fibers and quality of molded pieces was obtained in comparison with pure PHBV. Only in the case of wood–PHBV biocomposites was no significant improvement of properties obtained compared to biocomposites with other fillers used in this research. The use of natural fibers, in particular hemp fibers as a filler in the PHBV matrix, in most cases has a positive effect on improving the mechanical properties and quality of molded pieces. In addition, it should be remembered that the obtained biocomposites are of natural origin and are fully biodegradable, which are interesting and desirable properties that are a part of the current trend regarding the production and commercialization of modern biomaterials.

## 1. Introduction

In the current reality, an important problem is the topic of plastics waste management. Every year a huge amount of this type of waste increases in the world. Despite many recycling methods, not all plastics can be easily recycled. One alternative solution to this problem is the synthesis of biodegradable polymers or polymers produced from renewable raw materials. There are three groups of polymers satisfying these requirements [1,2]:biodegradable polymers made from petrochemical raw materials,non-biodegradable polymers made from renewable raw materials,biodegradable polymers made from renewable raw materials.

The third group of polymers combine the advantageous features of the first two groups. These are unique and the most desirable plastics in the current waste management problem.

Polyhydroxyalkanoates (PHAs) are a group of biodegradable polymers. They can be obtained by using microorganisms as well as from secondary raw materials and renewable agricultural sources [3]. It should be emphasized that PHA degrades without the production of toxic by-products [4].

PHB (polyhydroxybutyrate) is one of the polymers belonging to the group of polyhydroxyalkanoates. The first information about this polymer was introduced by Lemogine in the 1920s [5]. PHB has physical properties similar to polypropylene [6]; however, it is more brittle. The brittleness of PHB is associated with the formation of large spherulites [7], which may be due to the high purity of the obtained biopolymer. PHB is biodegradable and biocompatible, making it useful in tissue engineering and other biomedical applications [8].

Copolymerization of PHB with polyhydroxyvalerate (HV or PHV) allows (3-hydroxybutyrate-co-3-hydroxyvalerate) (PHBV) to be obtained and leads to a decrease of the degree of crystallinity of the polymer [9,10,11]. Compared to PHB, PHBV is characterized by reduced brittleness and stiffness, better tensile strength and greater elongation at break. By increasing the amount of HV in the polymer chain, the window of processing parameters can be improved [12].

Due to biodegradation to non-toxic compounds and easy processability, PHBV is still being modified and can be commercialized as the main substitute for non-biodegradable polymeric materials [13]. The similarity of some of its mechanical properties to polyolefins indicates that it can be a substitute for polymers from this group [14]. PHBV is the first biopolymer of the PHA group with properties that can be changed by controlling the content of the second monomer [15,16,17,18,19].

The possibilities of commercial application of this biopolymer are still difficult due to the narrow processing, relatively high brittleness and low flexibility, as well as significant production costs [4,20,21]. Future research plans must, therefore, assume the improvement of the mechanical properties and the extension of the processing window of biopolymers produced on the basis of PHB, including the PHBV biopolymer [22,23].

The use of natural and biodegradable plastics, such as PHBV, can contribute to the reduction of petrochemical plastic waste, which is not biodegradable and is very often landfilled and difficult to recycle. The work of Guo, Stuckey and Murphy [24] shows the possibility of developing a PHBV production system independent of the use of fossil fuels. PHBV polymers produced in the current production scale (2000 tons per year) have a slightly lower energy consumption during production per kg of polymer than in the case of petrochemical polymers. The current production processes and production scale of PHBV are still largely underdeveloped compared to the well-developed production of petrochemical polymers. One of the methods of extending the commercialization of green composites may be the use of natural fibrous fillers [25,26,27], especially in the PHBV matrix. As a result of their use, one should expect an improvement of the mechanical properties and a manufacturing cost reduction compared to pure biopolymer, while maintaining full biodegradability. The advantages of these fibrous fillers include the following [28,29,30,31,32]:The production costs are lower than in the case of synthetic fibers,Low density while maintaining satisfying strength and stiffness,The production process does not have a negative impact on the environment,Combustion/utilization of this type of waste does not generate toxic substances,Full renewable energy.

The main components of natural fibers of plant origin are cellulose, hemicellulose and lignin. The most common and available cellulose is contained in cotton (about 90% cellulose content) [33,34]. The amount of cellulose in plants can vary depending on the species and age of the plant. Although the chemical structure of cellulose from different natural fibers is the same, the degree of polymerization is different [34]. Hemicellulose, in turn, is a heterogeneous biopolymer of a group of polysaccharides that is less resistant to the action of diluted acids. Lignin, however, fills the spaces between the polysaccharide fibers, cementing them together. The presence of this fiber component stiffens the cell walls, protecting against chemical and physical damage [35]. The composition of cellulose, hemicellulose and lignin for individual plant fibers is presented in Table 1.

Polymers have different affinities for the fiber due to differences in their chemical structure. In order to increase the adhesion of fibers to the polymer matrix, reduce water absorption, increase the proportion of cellulose in the fiber and increase the degree of crystallinity, methods of fiber surface modification are used [36,37,38,39].

**Table 1 polymers-13-03934-t001:** Mass fraction of individual used plant fiber components (based on [34,40,41]).

Type of Fiber	Cellulose (% Mas.)	Hemicellulose (% Mas.)	Lignin (% Mas.)	Others (% Mas.)
Linen/Flax	71	18.6–20.6	2.2	1.5
Hemp	68	15	10	0.8
Deciduous trees	44 ± 3	32 ± 5	18 ± 4	0.2–0.8

The Young’s modulus and tensile strength of natural fibers such as sisal, jute, kenaf, hemp, flax are usually lower than for a glass fiber used in composites. The density of glass fiber is high, around 2500 kg/m^3^, while the density of natural fibers is much lower (about 1500 kg/m^3^). This is important when the mass of products made of composites becomes the key and therefore where this mass must be significantly reduced [42,43]. The properties of natural fibers depend on the cultivation conditions, harvest time and the method of processing and storage. So they are quite varied, which is quite a problem. In the work of Bos and co-authors [44] it was shown that hand-picked flax fibers have 20% higher mechanical properties than those harvested mechanically. In turn, the research in the work of Pickering and co-authors [45] shows that the fibers harvested after 5 days after the proper harvest time have lower strength properties, by as much as 15%. Of course, additional factors can affect the properties of the fibers. In the work of Charlet and co-authors, it has been noticed that Young’s modulus decreases with increasing humidity and grows with increasing temperature [46].

The mechanical properties of composites are in most cases improved by the addition of fibers to the polymer matrix. Therefore, the impact of fiber content on the strength properties of fiber reinforced composites is particularly interesting and important for many researchers [47,48,49,50].

A significant amount of research carried out with the use of PHBV composites with a matrix of plant fibers relates to the possibility of processing, focusing mainly on compression molding of thin films [14,51,52,53,54,55]. Both compression molding of thin composite sheets and processing on mini-extruders and mini-injection machines are limited processes due to the fact that they are difficult to relate to actual processing conditions. Therefore, in most cases, research results are difficult to transfer to a macro scale. The subjects of research by scientists are composites with a PHBV matrix filled with fibers of plant origin, most of which are coconut fibers [56,57,58,59,60], bamboo [14,61,62,63,64,65,66,67,68,69], abak [70,71], pineapple [56,72], sisal [73,74,75], agave [76,77], and vine shoots [78]

Due to the geographical location of Europe, the main sources of short cellulose fibers are wood, flax and hemp. When analyzing publications regarding the possibility of using short hemp, flax and wood fibers in the PHBV matrix, little information on the production, processing and properties of these types of composites was noticed [79,80,81]. The fibers used in the analyzed works were usually characterized by a fairly long length, with a very large dispersion of this value for a given set of fibers. This can lead to variable properties of composites. In addition, too long fibers can cause processing problems, especially during extrusion or injection molding. An important issue would be to consider the possibility of using powder of PHBV as the matrix and short fibers with a length of 1 mm and a very small length distribution. This would probably allow composites to be obtained with a higher degree of homogenization. It should be mentioned that in some works no conventional extruders were used, hence the observations and obtained research results may not reflect possible phenomena and problems on an industrial scale. This would also reduce the content of the expensive matrix in the biocomposite in order to reduce the producing costs of composite.

The aim of the study was to assess the impact of the type of filler on the processing and mechanical and functional properties of biocomposites with the PHBV matrix—such comprehensive work on analyzed fillers and matrices has not been found in the literature. The fillers tested, i.e., wood, flax and hemp fibers, were selected due to their availability and popularity in the authors’ geographical area of residence (Europe). No work was found comparing these three types of composites in terms of production, processing and evaluation of mechanical, processing and functional properties. The results of the research may have a scientific as well as application aspect, allowing us to indicate the possibility of manufacturing utility products in the injection molding process of the manufactured composites, especially in Europe, where these fillers are easily available and popular.

## 2. Materials and Methods

### 2.1. Materials

As the polymer matrix, PHBV with the Enmat Y1000 trade name of Helian Polymers (Belfeld, The Netherlands) in powder form was used. The molar content of HV in the biopolymer was 8%, the density of the biopolymer was 1250 kg/m^3^ and the softening point was in the range from 165 to 175 °C [82].

As fillers in the polymer matrix fibers of plant origin, the following were used:Wood fibers,Hemp fibers,Flax fibers.

Wood fibers with the trade name Lignocel C120 were about 1 mm long. Hemp and flax fibers were supplied by EKOTEX company (Kowalowice, Poland) and were characterized too by the length of 1 mm. The average ratio of length to diameter of all fibers (L/d) was about 10 (for L = 1 mm). Three types of composites with a variable type of filler were prepared, where their mass content in the polymer matrix was 30%.

### 2.2. Investigation of Fibers

A HITACHI S-3400 scanning electron microscope (SEM) produced by Hitachi Inc. (Tokyo, Japan) was used to carry out the test of the microstructure of the fibers. In the case of hemp fibers the rectilinear geometry of fibers can be seen in the SEM photographs analyzed (Figure 1). The surface of hemp fibers is slightly more developed than flax fibers, which causes undoubtedly better fiber adhesion to the matrix. The diameter of flax fiber is noticeably smaller than the diameter of hemp fiber. The smaller diameter of flax fibers is visible and they are also more twisted. It should also be noted that wood fibers have irregular size and geometry, and the surface of the fibers is the most developed among the three analyzed fillers.

### 2.3. Manufacturing of Biocomposites

In order to minimize the presence of moisture/water in the obtained mixtures (PHBV–flax fibers, PHBV–hemp fibers, PHBV–wood fibers), they were dried before the extrusion process. Both hemp, flax and wood fibers as well as PHBV biopolymer tend to absorb water, which may have a negative impact on the possibility of obtaining a homogeneous structure of the composite fiber fractions, and PHBV with a high degree of water absorption may tend to clump into larger agglomerates. Moreover, drying should be performed in order to reduce the occurrence of possible air bubbles in the cross-section of the obtained extrudate. Drying was carried out in a Chemland laboratory dryer, model DZ-2BC (produced by Chemland company, Szczecin Stargard, Poland) with a maximum power of 1400 W and a capacity of 52 L, equipped with a Value pressure pump model V-i120SV with a flow of 51 L/min. The mixtures were dried at a temperature of 90 °C for 6 h with a vacuum in the drying chamber of 0.02 MPa.

Biopolymer and biocomposites were extruded using the ZAMAK EHP-25E single screw extruder (produced by ZAMAK Mercator company, Skawina, Poland) at a constant temperature in individual extruder heating zones. The temperature ranged from 145 °C (Zone 1) to 160 °C (Head) for pure PHBV and from 150 °C (Zone 1) to 170 °C (Head) for biocomposites. The temperature increase during biocomposite extrusion resulted from the higher viscosity of the extrudate obtained.

A summary of the set temperatures is shown in Table 2. The extrusion was carried out using an extrusion granulation station equipped with a cooling bath and a granulator. All materials were extruded at a constant screw speed of 100 rpm. The choice of such a rotational speed of the screw was justified by the fact that in the case of a higher speed, the process was unstable, i.e., the fiber accumulated between the cylinder and the extruder head, and the outgoing extrudate was characterized by very low viscosity. In turn, reducing the screw speed below 100 rpm resulted in degradation of the biocomposite in the plasticizing system due to too long of a heating time. The granules obtained were used to manufacture specimens of a dogbone shape intended for testing the mechanical properties.

### 2.4. Manufacturing of Samples

The Dr. Boy 55E injection molding machine (produced by BOY Machines Inc. (Exton, PA, USA)) equipped with a Priamus data acquisition and processing system (by Priamus System Technology, Rheinweg, Switzerland) for control and monitoring of the injection molding process was used. In the study, an injection mold with special inserts designed for uniaxial tensile sample tests (in accordance with EN ISO 527-1 [83] was used. The samples of dogbone geometry for all types of biocomposites were produced at the same processing parameters shown in Table 3. Only during the production of samples of biopolymer were lower mold and melt temperatures (Table 3) to obtain a higher viscosity of polymer. For instance, the specimen from a biocomposite filled with flax fibers is shown in Figure 2. The resulting samples were used to test the mechanical properties and the quality of the molded piece (specimens).

Using the same parameters for all biocomposites, an increase in the maximum pressure values (up to approx. 25 MPa) in the mold cavity was observed (Figure 3) after adding the flax and hemp filler to the PHBV matrix. In the case of injection of pure PHBV and PHBV wood–fiber biocomposite, the fastest solidification of the material was noticeable, which indicated the possibility of shortening the pressure time. In addition, significantly lower pressure values were observed for wood fiber–PHBV biocomposite than for other biocomposites (maximum pressure was approx. 13 MPa).

### 2.5. Testing Methods

The tests of the obtained biocomposites were carried out for a package of seven samples from each composite and biopolymer.

The shrinkage properties of composites were tested based on the EN ISO 294-4 [84] standard. The degree of water absorption was tested based on the EN ISO 62 [85] standard. Microstructure tests were carried out using a HITACHI S-3400 scanning electron microscope (SEM) based on specimens from the uniaxial tensile test.

In order to determine the strength properties, a Z030 testing machine manufactured by Zwick Roell (Ulm, Germany) was used. The uniaxial tensile test was performed in accordance with EN ISO 527-1. The statistical analysis was performed. In the evaluation of the strength properties, the following were taken into account: tensile strength (σM), relative elongation at maximum tensile strength (ε_M_) and Young’s modulus (E). The results were analyzed by determining the arithmetic mean (AM), the standard deviation (s) and the coefficient of variation (V).

The hardness tests were carried out using the Brinell method in accordance with the EN ISO 2039-1 standard in two areas of the sample (Figure 4), i.e., in the measuring zone (zone A) and in the grip zone (zone B). A Zwick 3106 hardness tester manufactured by Zwick Roell (Ulm, Germany) was used for this purpose.

Biocomposite samples were also tested by means of the impact tensile test. The tensile impact strength test was determined in accordance with EN ISO 8256 [86]. The CAEST 9050 impact pendulum hammer produced by Instron Inc. Europe (Buckinghamshire, UK) was used for this purpose. In order to carry out the correct test, some samples were prepared from the dogbones samples intended for the uniaxial tensile test. Their geometry was modified according to the standard. The notch was milled for the entire sample packages.

## 3. Results

### 3.1. Determination of Shrinkage

The longitudinal, transverse and thickness shrinkage were determined (Figure 5). It was observed that the use of examined types of fillers in most cases had a positive effect on reducing the shrinkage of product compared to the pure biopolymer.

### 3.2. Water Absorption Assessment

A water absorption study was carried out. The expected fact was an increase of water absorption after adding cellulose fillers to the polymer matrix (Figure 6). This may be due to the fact that hydroxyl (OH) groups in cellulose, hemicellulose and lignin build up a large amount of hydrogen bonds inside the macromolecule and between macromolecules in the cell wall of plant fibers. The action of water on plant fibers breaks these bonds. The hydroxyl groups then form new hydrogen bonds with water molecules that promote fiber swelling and, consequently, increase the mass of the biocomposite.

### 3.3. Uniaxial Tensile Test

Representative stress–strain characteristics from the uniaxial tensile test are shown in Figure 7. When analyzing the results (Table 4), it was noted that the use of flax and hemp filler improved the strength properties of biocomposites compared to the pure biopolymer.

### 3.4. Tensile Impact Strength Test

Higher values of impact tensile strength of biocomposites with flax and hemp fiber were noted (Figure 8), where an increase of approx. 62% was noted, compared to pure biopolymer. In addition, it was noted that the use of wood fiber as a filler in the PHBV matrix did not improve the tensile impact strength compared to the pure biopolymer.

### 3.5. Brinell Hardness Test

Brinell hardness tests for biocomposites were carried out in two areas of the samples dedicated for the uniaxial tensile test, i.e., in the neck (area A) and in the grip (area B) of every sample. The test results are shown in Figure 9.

### 3.6. Study of Microstructure

By analyzing SEM photographs (Figure 10) of biocomposite sample fractures after the uniaxial tensile test, it could be seen that in biocomposites reinforced with hemp fibers, the fibers cracked transversely to their length. In addition, hemp fibers had no tendency to delaminate and had a rectilinear geometry. In the case of flax fibers, it could be seen that the fibers located in the matrix had a smaller diameter than hemp fibers and were mostly delaminated and more twisted. In the case of wood fibers, cracking at the fiber–matrix boundary was noticeable due to the visible free space at the boundary of the fiber located in the biopolymer. This may indicate low adhesion and may result in worse mechanical properties compared to other biocomposites.

In addition, wood fibers had irregular geometry and size—it was difficult to observe a constant length/diameter ratio (L/d). In turn, similar values of the ratio of length to diameter of the fibers could be observed in the case of flax and hemp fibers located in a polymer matrix. It should also be noted that the dispersion of wood fibers in the matrix was quite low—this fact may also be evidence of worse strength properties of the biocomposite. For biocomposites with flax and hemp fibers, the degree of fibers dispersion in the polymer matrix was more regular than in composite with wood fiber.

## 4. Discussion

Over the last 30 years, a continuous and accelerating process of introducing biodegradable materials into industrial use has been observed. Environmental protection requirements, rising costs of petroleum products and striving to reduce the carbon footprint of civilization are driving researchers towards materials of organic origin. Particular interest is directed towards composite materials, in which, thanks to the combination of various properties of the base material and reinforcement, the possibility of a wide and varied shaping of functional features is gained. Introducing new materials into common use requires one not only to demonstrate the immediate properties of this material, but also to analyze the processes taking place during their long-term use in various environmental conditions. It is necessary to learn the rules governing the phenomena related to the process of producing the material itself.

In the analyzed works, Keller, for example, discussed the influence of the hemp fiber production process, and thus the length of fibers, on their mixing with the matrix (PEA or PHBV) and the resulting differences in the basic mechanical properties of the obtained composites [80]. He obtained a composite material with a fiber volume fraction of 32%. The use of hemp fibers was characterized by a length of 5 to 25 mm, which resulted in a problem with the extrusion of the biocomposite. A heterogeneous granulate was obtained; hence, some attempts were made to spin the threads for the extrusion process. During the extrusion process, shortening of the fibers and loss of rectilinear geometry were also noted. When assessing the test results, it was found that by adding hemp fibers to the PHBV matrix, the tensile strength did not increase, but the maximum elongation of the sample was reduced compared to pure PHBV.

The study of the influence of the fillers (in a form of: cellulose, jute, abaca fibers) on the properties of composites taking into account various matrix materials (PLA, PHBV) and their comparison with mixtures based on PP can be found in [22]. The tested composite materials were characterized by comparable or even better properties than composites with a PP matrix. There was an increase in rigidity, but also of the strength and notched impact strength.

A wider scope of research on the influence of the reinforcement content in the composite (wood flour/PHBV) is described in [81]. The authors focus on the influence of temperature on the properties of the obtained mixture, both in the process of its extrusion and in the finished product. When analyzing the environmental factors affecting biodegradable composites, it is also impossible to ignore the time of this impact and the type of factor. When analyzing the results, it was noticed that increasing the content of wood flour causes a significant increase in the tensile modulus (it was improved by 167% compared to the pure PHBV). The tensile strength decreased significantly with the increase in the content of wood flour.

The problem of changing the mechanical properties of structures (flax/PHBV) was also described [20]. The work focused on the evaluation of the properties of PHBV–flax fiber composites with variable flax fiber content for samples produced by injection molding and compression pressing. A comparative analysis of the mechanical properties of the obtained composites with the mass fraction of filler from 10 to 30% was performed. Similar Young’s modulus values were reported for the samples produced by the two methods mentioned. Slightly higher values of Young’s modulus were obtained for injection-molded samples and amounted to approx. 6 GPa (30% by volume of the filler content).

The use of fillers (not only of plant origin) in the polymer matrix reduces the shrinkage of the composites. The geometry of the fibers (L/d) especially reduces the longitudinal shrinkage because during the flow of the plasticized polymer composite in the channels feeding to the forming cavity, the fibers are arranged along the flow direction, which results in the formation of the polymer matrix reinforcement counteracting the shrinkage of the composite in the longitudinal direction [87,88,89,90]. The lowest value of longitudinal shrinkage was obtained for a biocomposite with hemp fibers—the value of longitudinal shrinkage compared to pure biopolymer was reduced by about 74%. In addition, the lowest values of transverse and in thickness shrinkage were also observed for biocomposite with hemp fiber—an approx. 37% decrease in transverse shrinkage and an approx. 44% decrease in the thickness shrinkage compared to the pure biopolymer. The smallest decrease of shrinkage compared to pure PHBV was observed for the PHBV–wood fiber biocomposite. A 56% decrease of longitudinal shrinkage value was then observed, and in the event of the shrinkage in thickness, there was a 26% decrease related to pure PHBV. In addition, there was no change in transverse shrinkage relative to PHBV for the wood fiber–PHBV biocomposite.

In the case of the uniaxial tension test made for a biocomposite with hemp fiber, an approx. 167% increase in Young’s modulus value and an approx. 21% increase in tensile strength were noticed. On the other hand, for a biocomposite filled with flax fiber, there was an approx. 156% increase in Young’s modulus and an approx. 13% increase of tensile strength. In the case of using wood fiber, an approximate 133% increase in the Young’s modulus was noted, and an approximate 14% decrease in the tensile strength. For all types of plant-based fillers used in the PHBV matrix, a decrease in relative elongation at ultimate tensile strength was noted, where the biggest drop was noted for wood-filled biocomposite and was about 73% less compared to pure PHBV

Higher values of hardness in the area A (Figure 9) were observed for all biocomposites (for wood fiber—an increase of approx. 46%, for flax fiber—an increase of approx. 34% and for hemp fiber—an increase of approx. 36%) compared to pure PHBV. In turn, analyzing area B (Figure 9), a smaller increase of the hardness for a biocomposite with wood fiber (36%), flax (21%) and hemp (24%) compared to a pure biopolymer was observed. A greater resulting dispersion performed in B area was also noted compared to area A of the sample.

As can be seen in the water absorption diagram, the difference in water absorption between the produced composites was small. This is due to the similar morphology of plant fibers (Table 1). It should be noted, however, that there was a significant increase in the water absorption of composites compared to pure PHBV. Hydroxyl groups (OH) in cellulose, hemicellulose and lignin build a large amount of hydrogen bonds inside the macromolecule and between macromolecules in the cell wall of plant fibers. The action of water on plant fibers causes these bonds to break. The hydroxyl groups then form new hydrogen bonds with the water molecules, which cause the fiber to swell. The swelling of the cell wall generates very high forces. The theoretical value of the pressure may be around 165 MPa [91,92,93]. Cellulose fibers interact with water not only on the surface but also in the entire volume. The structure of cellulose-based materials consists of crystalline and amorphous regions. Amorphous regions readily absorb chemical compounds such as dyes and resins, and the presence of crystalline regions makes chemical penetration difficult [94]. The possibility of water absorption by cellulose fibers depends on the following, among others: (a) cellulose purity: crude cellulosic material, such as unwashed sisal fibers, absorbs at least twice as much water as washed fibers due to the 24% pectin content; (b) degree of crystallinity: all the OH groups in the amorphous phase are accessible to water, while only a small amount of water interacts with the surface OH groups of the crystalline phase. The main disadvantage of cellulose fibers is their highly polar nature, which makes them incompatible with non-polar polymers. The poor resistance to water absorption makes the use of natural fibers less attractive for products that will be used under the influence of external factors (e.g., rain, snow, hail) [33,34,95].

Solle and co-authors researched the biodegradability of flax/PHBV untoughening and toughening composites. Composites containing 30% filler were prepared by pressure pressing PHBV powder interspersed with unidirectional flax fabric. In the natural soil environment, biodegradation was carried out. Biodegradability was assessed by weight loss analysis, optical microscopy and electron microscopy. The biodegradability of the composite was significantly increased by the addition of flax fibers as compared to pure PHBV. Toughened composites showed a faster degradation rate than untoughened ones [96].

Zaidi and Crosky researched PHBV reinforced with a unidirectional flax fabric. The addition of flax resulted in a 4-fold increase in tensile properties, a 3-fold increase in flexural properties and a 20-fold increase in impact properties with minimal change in thermal properties. Moreover, they found that unidirectional flax PHBV reinforcement significantly extends the range of PHBV applications in terms of mechanical properties [97].

Mazur and Kuciel indicated, apart from the issues of aging, the relationship between temperature and mechanical properties of the tested compositions. Biodegradable composites based on PHBV reinforced with 7.5% or 15% by weight of wood fibers were produced by injection molding. The obtained composites were characterized by an increase in the value of Young’s modulus, but also a decrease in strength and impact properties. A comparative analysis of the experimentally measured values of Young’s modulus with the values obtained in various theoretical micromechanical models was performed. The Haplin–Kardas model was found to be similar to the experimental data. Biodegradation studies of biocomposites in physiological saline solution at 40 ◦C were also carried out in order to investigate the loss of mass. It was observed that the presence of fibers influences the water absorption rate, and the highest index was observed for composites with 15% by weight of filler [98].

Taking into account the microstructure test, it can be noted that in the case of three types of biocomposites, a greater porosity of the structure in relation to pure PHBV is visible. There is also a visible lack of significant difference in the geometry and surface quality of the fibers located in the PHBV matrix in relation to the SEM photography of individual fibers (Figure 1). This may be some evidence of well-chosen processing parameters for biocomposites, with no degradation of the fibers.

## 5. Conclusions

Three types of biocomposites with PHBV matrix filled with hemp, wood and flax fibers were produced in the extrusion process. In the case of the processing of the biocomposites filled with flax and hemp fiber, higher pressure values in the mold cavity were obtained compared to pure PHBV. For biocomposites filled with wood fiber, pressure values in the mold cavity, similar to pure PHBV, were achieved.

In the case of analyzed biocomposites, there was an improvement in both their mechanical properties and an improvement in the quality of products made of them, especially for composites with flax and hemp fibers. The best results were obtained for a biocomposite filled with hemp fibers. In comparison to PHBV, an approx. 167% increase in the Young’s modulus was found, approx. 21% increase in the tensile strength value and approx. 62% increase in tensile strength. For composites with wood fibers, much worse mechanical properties were obtained compared to other biocomposites and pure PHBV. In the case of the shrinkage value of the moldings, a significant reduction in shrinkage, in particular longitudinal shrinkage, was found for all biocomposites. It should be noted that the value of longitudinal shrinkage compared to the pure biopolymer was reduced by up to approx. 74% for a biocomposite filled with hemp fibers. For all biocomposites with a filler of plant origin, greater water absorption was observed compared to pure biopolymer.

The use of natural fibers, in particular hemp fibers as a filler in the PHBV matrix, in most cases has a positive effect on improving the mechanical properties and quality of molded pieces. In addition, it should be remembered that biocomposites obtained are of natural origin and are fully biodegradable, which is interesting and desirable for properties that are part of the current trend regarding the production and commercialization of modern biomaterials. The directions of searching for the possibility of using these materials focus on the production of consumer products that would meet the following criteria: wear out after a certain period of use; they are not repaired when damaged; they can be loaded during use; they can come into direct contact with living organisms. Ultimately, taking into account these criteria, the manufactured biocomposites may apply, inter alia, in the production of plastic pallets, fruit/vegetable boxes, containers for hospital waste such as gauze pads, etc., elements securing electronic products in cardboard boxes and packaging. The companies producing such products, e.g., in injection molding processes, may be the recipient of such a solution.

## Figures and Tables

**Figure 1 polymers-13-03934-f001:**
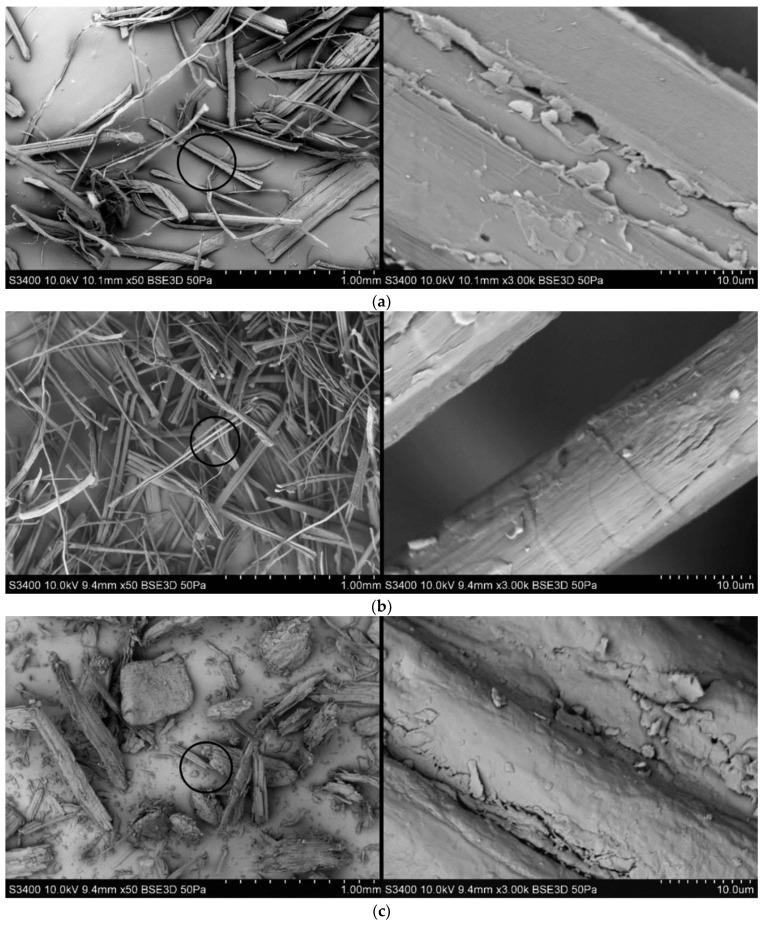
SEM photography of (**a**) hemp, (**b**) flax and (**c**) wood fibers.

**Figure 2 polymers-13-03934-f002:**
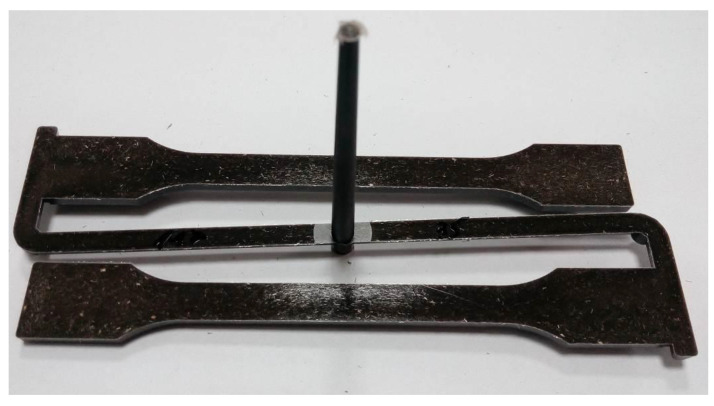
Samples made of PHBV–flax fiber biocomposite (30 wt.%).

**Figure 3 polymers-13-03934-f003:**
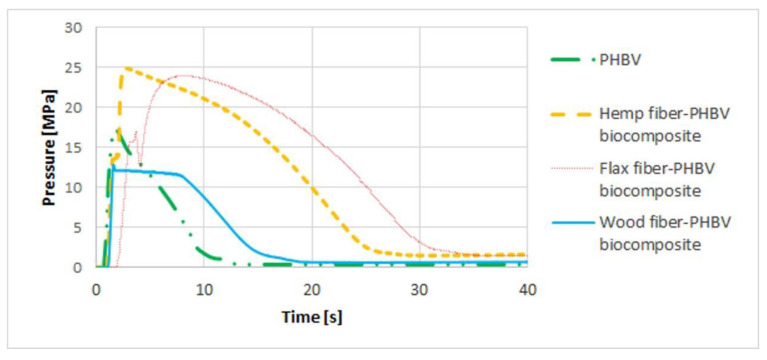
The pressure profile in mold cavity for biocomposites with a variable type of filler and pure biopolymer.

**Figure 4 polymers-13-03934-f004:**
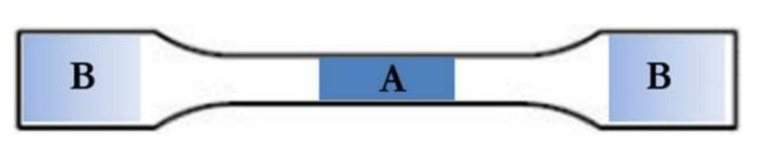
The measured areas of specimens for hardness tests.

**Figure 5 polymers-13-03934-f005:**
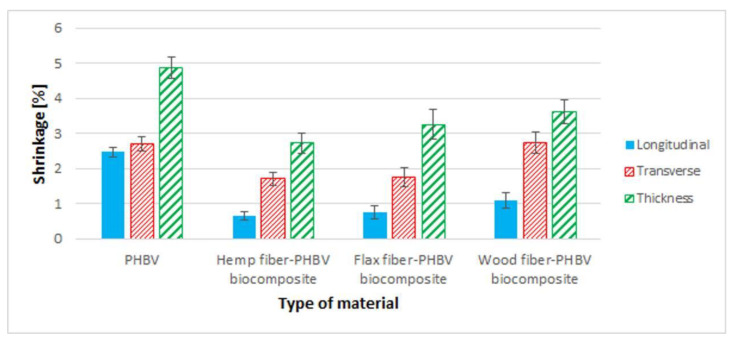
Shrinkage: longitudinal and transverse, in thickness for biocomposites with a variable type of filler and for pure PHBV biopolymer.

**Figure 6 polymers-13-03934-f006:**
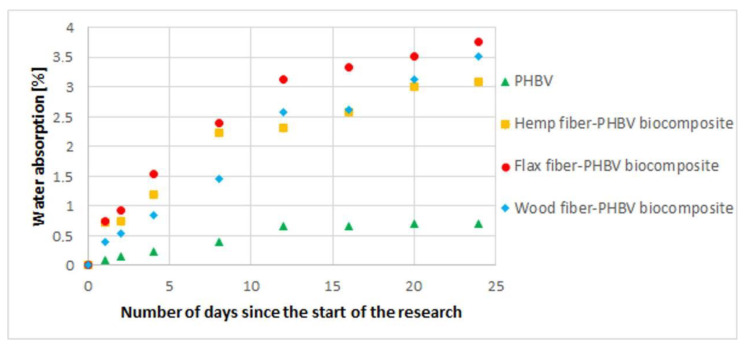
Water absorption for biocomposites with a variable type of filler and for pure PHBV.

**Figure 7 polymers-13-03934-f007:**
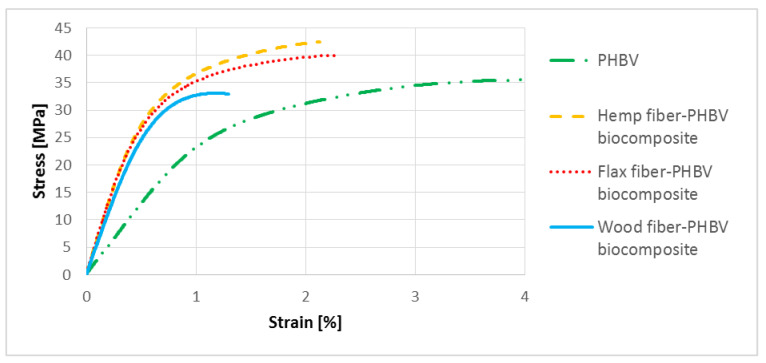
Stress–strain characteristics for pure PHBV and biocomposites with different types of filler.

**Figure 8 polymers-13-03934-f008:**
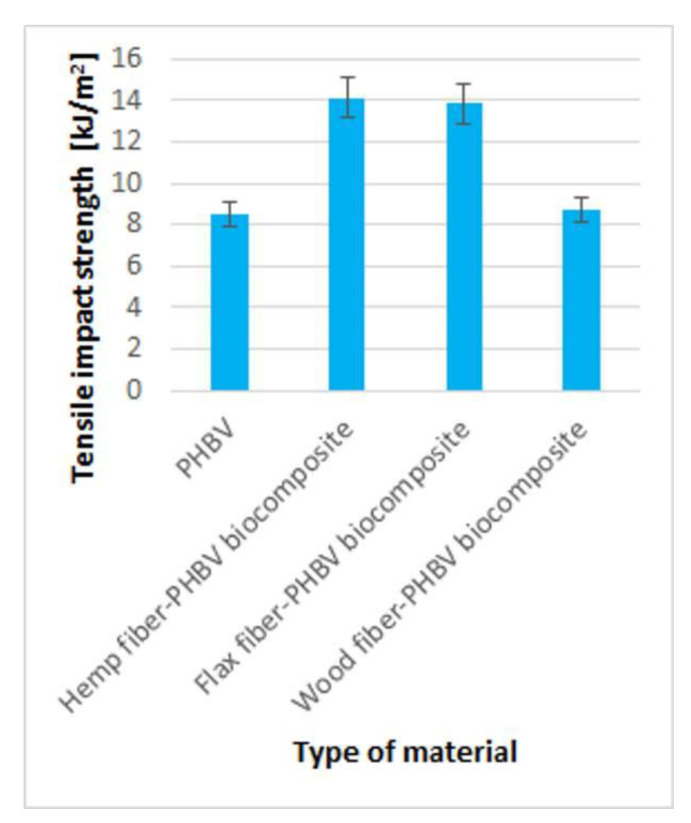
The results of tensile impact strength test for pure polymers and biocomposites with various fillers type.

**Figure 9 polymers-13-03934-f009:**
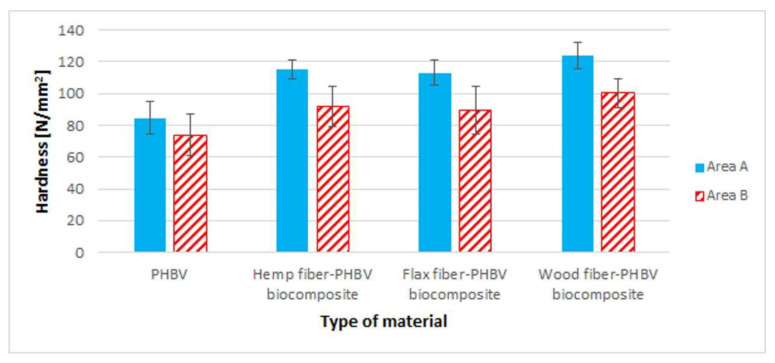
Hardness for pure polymers and biocomposites with various filler types for A and B areas.

**Figure 10 polymers-13-03934-f010:**
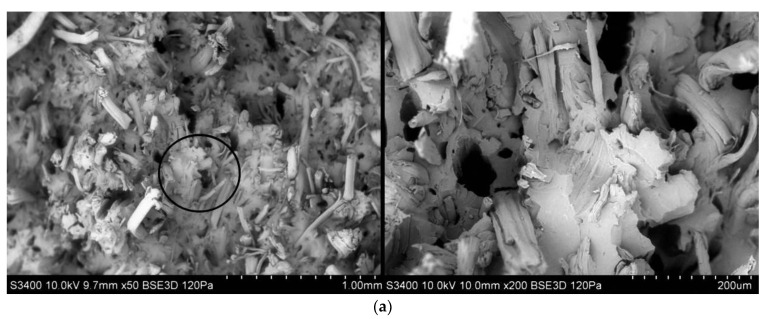
SEM photographs of fractures for biocomposites samples filled with fibers of (**a**) hemp, (**b**) flax and (**c**) wood; (**d**) SEM photograph for pure PHBV.

**Table 2 polymers-13-03934-t002:** The heating zone temperatures of a single screw extruder.

Type of Material	Temperatures (°C)				
	Head	Zone 3	Zone 2	Zone 1	Feed Hopper Zone
PHBV	160	160	155	145	50
biocomposites	175	170	160	150	35

**Table 3 polymers-13-03934-t003:** The sample processing parameters of the injection molding process.

Parameter	PHBV	Biocomposites
Mold temperature (°C)	60	85
Melt temperature (°C)	167	185
Cooling time (s)	25	25
Packing time (s)	25	25
Packing pressure (MPa)	30	30
Flow rate (cm^3^/s)	35	35

**Table 4 polymers-13-03934-t004:** The results from uniaxial tensile test for pure polymers and biocomposites with different filler types.

Type of Material	Statistics	E (MPa)	σ_M_ (MPa)	ε_M_ (%)
PHBV	$\bar{x}$	2617.37	35.48	4.12
	s	112.02	0.86	0.15
	V	4.28	2.42	3.63
Hemp fiber–PHBV biocomposite	$\bar{x}$	6992.31	42.90	2.28
	s	199.44	0.71	0.06
	V	2.85	1.65	2.60
Flax fiber–PHBV biocomposite	$\bar{x}$	6701.86	40.18	2.50
	s	216.86	0.40	0.25
	V	3.24	0.99	9.94
Wood fiber–PHBV biocomposite	$\bar{x}$	6110.16	30.68	1.13
	s	362.87	0.79	0.06
	V	5.94	2.57	5.23

## Data Availability

Not applicable.

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
