# Peer review of "The Influence of Chosen Plant Fillers in PHBV Composites on the Processing Conditions, Mechanical Properties and Quality of Molded Pieces"

_polymers, 2021, doi:10.3390/polym13223934_

Round 1

Reviewer 1 Report

The manuscript (polymers-1422541) presents the fabrication of some novel biocomposites. However, the overall logic of the manuscript is insufficient, so the reviewer recommends that the manuscript should be major revised. There are some issues that should be addressed before it can be accepted:

1) The overall logic of the manuscript is insufficient. The innovations and improvements compared to previous researches haven't been
clarified clearly.

2) The results given are insufficient, not explained deeply. The purpose of the work is not clear enough, it made me think that the final application is aimed for packaging but it is unclear. I suggest additional characterizations on thermal, flexural and viscoelastic properties. If the end-use is aimed for packaging, it will be better to show that these biocomposites can be formed into films. 

3. The quality of English leads to a poor readability of the manuscript. There are lots of grammar mistakes, such as "a various types" on line 49, "It should be emphasize that" on line 51, the unity of lines 87, 88 with 89, 91 and many more. The manuscript needs intense improvement in English and level of sentences. 

4. There are some mistakes with format. In kg/m3, 3 must be written as superscript, line 103, 104. The degree symbol should be uniform throughout the text. Please check the manuscript carefully and correct all the errors.

Author Response

Dear Sir/Madam
Please find attached answers to questions / suggestions.
Kind regards,
Wiesław Frącz 

Reviewer 2 Report

The manuscript “Influence Of Chosen Plant Fillers In PHBV Composites On The Processing Conditions, Mechanical Properties And Quality Of Molded Pieces” fits well in the scope of Polymers and especially of the special issue “Polymers from Renewable Sources and Their Mechanical Reinforcement”. The study is well-designed, however the novelty and the contribution of this work in the field of biocomposites must be explained better, since there is a lot of bibliography on the topic. Furthermore, there is a lack of explanation of the obtained results, since the reasons behind the differences between the different fillers as well as the reasons why mechanical properties were enhanced are not hypothesized. Improvements are neccesary before publication, as described below. General comments 1. The use of English language must be significantly improved. 2. The novelty of the work is not adequately described, as the mechanical properties of PHBV biocomposites have been studied multiple times and previous work done on this topic is in my opinion. 3. The results are presented in the form of a report rather than explained. The effect of the different types of fillers must be evaluated. The reasons why mechanical properties were improved, which is in contrast with some of the mentioned papers in the discussion, must be identified. Specific comments 4. Introduction, line 43: Is there a formal definition of green polymers that described biodegradable and biobased polymers? If so, it must be cited. If not, the definition should be removed, as the term “green” usually has several different interpretations. The terms “Green composites” or “biocomposites” are more common, and they refer to biodegradable polymer matrices with natural fillers. 5. Introduction, line 44: The statement that biodegradable and biobased polymers are the most desirable is not exactly accurate – biodegradation is required in specific applications, and they are not suitable for long term applications, where a non-degradable biobased polymer would be more appropriate. Where could the produced materials find applications? 6. Introduction: what are the applications of PHB, its production capacities and their role in the transition to a circular plastics economy? 7. Introduction: There is a lot of work available on both wood/biodegradable plastic composites as well as natural fibers/ biodegradable plastic composites in the literature (e.g. 10.1016/j.indcrop.2014.08.034, 10.1177/0892705718816354, 10.1515/secm-2016-0072). Also with PHBV matrix, even when the fillers are not identical, most bast fibers have similar compositions and properties, therefore it must be mentioned (e.g. 10.1007/s10924-009-0127-x, 10.1002/app.50182, 10.1016/j.compscitech.2012.01.021, 10.1007/s10924-020-01884-8, 10.1007/s10924-020-01979-2, 10.3389/fmats.2019.00275, 10.1016/S0266-3538(03)00102-7, 10.3390/su11082411, 10.1002/adv.21789, 10.3390/molecules24193538, 10.1016/B978-1-78242-373-7.00008-1, 10.1007/s10924-020-01781-0, 10.1016/j.compositesa.2019.05.009, 10.1002/9783527820078.ch8). It is mentioned in the discussion part briefly but not in the introduction. 8. Introduction: another very important disadvantage of these natural fillers is that their hydrophilic nature prevents them from having strong interfacial interactions that ultimately result in gaps and holes and premature mechanical failure. A lot of work has been conducted on functionalization and compatibilizers to improve their properties that must be at least mentioned (e.g. https://doi.org/10.3390/ma9040303 and work of the authors 10.3390/polym13121965, 10.12913/22998624/135399) 9. What is the difference of the PHBV hemp composites of this work with the previous work of the authors (10.12913/22998624/135399) 10. Why was 30 wt% filler content chosen? 11. Section 2.1: What is the composition of the fibers used? 12. Section 2.3: Natural fibers are very hydrophilic; where they dried before extrusion? 13. Section 3.1: Was there a significant effect of the fiber type on shrinkage? Where do these results stands in comparison with other data from the literature if available? Why did the presence of fibers reduce shrinking? 14. Section 3.2: Was there a significant effect of the fiber type on water adsorption and why? 15. Section 3.3: Was the difference in mechanical properties in the presence of the fillers statistically significant? 16. Section 4: Why was the properties in this work improved and what was the overall effect of the different filler types? The obtained results must be explained considering the chemistry of the materials as well as the data obtained from previous work reported in the literature. 17. The conclusions section must be shortened significantly to include only the most important information and provide insight as to why these results were obtained. 18. Please remove the work “the” on the titles of the sections. Finally, I would like to thank the authors for considering my comments and wish them the best for their future research.

Author Response

Dear Sir/Madam
Please find attached answers to questions / suggestions.

The manuscript “Influence Of Chosen Plant Fillers In PHBV Composites On The Processing Conditions, Mechanical Properties And Quality Of Molded Pieces” fits well in the scope of Polymers and especially of the special issue “Polymers from Renewable Sources and Their Mechanical Reinforcement”.

The study is well-designed, however the novelty and the contribution of this work in the field of biocomposites must be explained better, since there is a lot of bibliography on the topic. Furthermore, there is a lack of explanation of the obtained results, since the reasons behind the differences between the different fillers as well as the reasons why mechanical properties were enhanced are not hypothesized.

The above-mentioned note has been improved and expanded in the Introduction and Discussion chapters.

Improvements are neccesary before publication, as described below.

General comments

  1. The use of English language must be significantly improved.

Thank you for your attention. The entire manuscript has been revised by a certified English teacher. The current version of the work should now be linguistically correct.

  1. The novelty of the work is not adequately described, as the mechanical properties of PHBV biocomposites have been studied multiple times and previous work done on this topic is in my opinion.

The aim of the study was to assess the impact of the type of filler on the processing, mechanical and functional properties of biocomposites with the PHBV matrix - such comprehensive work on the above-mentioned fillers and matrix have not been found in the literature. The fillers tested, i.e. wood, flax and hemp fibers, were selected due to their availability and popularity in the authors' geographical area of residence (Europe). No work was found comparing these three types of composites in terms of production, processing and evaluation of mechanical, processing and functional properties. The results of the research may have a scientific as well as application aspect, allowing to indicate the possibility of manufacturing utility products in the injection molding process of the above-mentioned composites, especially in Europe, where these fillers are easily available and popular.

  1. The results are presented in the form of a report rather than explained. The effect of the different types of fillers must be evaluated. The reasons why mechanical properties were improved, which is in contrast with some of the mentioned papers in the discussion, must be identified. Specific comments

Thank you for your attention. We tried to improve it and expand it in the discussion.

  1. Introduction, line 43: Is there a formal definition of green polymers that described biodegradable and biobased polymers? If so, it must be cited. If not, the definition should be removed, as the term “green” usually has several different interpretations. The terms “Green composites” or “biocomposites” are more common, and they refer to biodegradable polymer matrices with natural fillers.

Thank you for your attention - the actual phrase "green polymers" is rarely used in the literature and interpreted differently [Hatakeyama, T., & Hatakeyama, H. (2006). Thermal properties of green polymers and biocomposites (Vol. 4). Springer Science & Business Media .; Khalaf, M. N. (2016). Green polymers and environmental pollution control. CRC Press., Gomez, J. G., Méndez, B. S., Nikel, P. I., Pettinari, M. J., Prieto, M. A., & Silva, L. F. (2012). Making green polymers even greener: towards sustainable production of polyhydroxyalkanoates from agroindustrial by-products. Advances in applied biotechnology, 41-62., 9. Scott, G. Green'polymers. Polymer degradation and stability, 2000, 68, 1-7.] Therefore this phrase has been removed from the work.

  1. Introduction, line 44: The statement that biodegradable and biobased polymers are the most desirable is not exactly accurate – biodegradation is required in specific applications, and they are not suitable for long term applications, where a non-degradable biobased polymer would be more appropriate. Where could the produced materials find applications?

The produced biocomposites can be used in the details of a specific purpose. Directions of searching for the possibility of applying the above-mentioned materials, they focused on the production of a utility product that would meet the following criteria:

  • it wears out after a certain time of use
  • it cannot be repaired after damage
  • it can function as a product that is loaded during use
  • it can come into direct contact with living organisms.

Ultimately, taking into account the above-mentioned criteria these biocomposites may apply, inter alia, in the production of: plastic pallets, fruit / vegetable boxes, containers for hospital waste such as gauze pads, etc., elements securing the electronics sold in cardboard boxes and packaging. The recipient of the solution may be companies producing details of the above-mentioned intended use in the injection molding process. It should be noted that the composites described in the paper have been submitted to the office for patent protection.

  1. Introduction: what are the applications of PHB, its production capacities and their role in the transition to a circular plastics economy?

The lamellar structure of PHB provides excellent gas barrier properties with water vapor permeability, making it suitable for low cost food packaging applications. PHB is biodegradable and biocompatible, making it useful in tissue engineering and other biomedical applications such as surgical sutures, thermogels as carriers for drug delivery with controlled release, surgical meshes, wound dressings and absorbable nerve guides, tissue scaffolds for regeneration bones and nerves, cardiovascular and cartilage support [https://doi.org/10.1016/j.msec.2017.11.006]. The use of natural and biodegradable plastics, such as PHB, can contribute to the reduction of petrochemical plastic waste, which, as we know, is not biodegradable and is very often stored and difficult to recycle.

In the work of Guo, Stuckey and Murphy [Guo M., Stuckey D.C., Murphy R.J .: Is it possible to develop biopolymer production sys-tems independent of fossil fuels? Case study in energy profiling of polyhydroxybutyrate-valerate (PHBV). Green Chemistry, 15 (2013), 706-717.] Presents the possibility of developing a PHBV production system regardless of the use of fossil fuels. PHBV polymers produced in the current production scale (2000 tons per year) have a slightly lower energy consumption during production per kg of polymer than in the case of petrochemical polymers. The current production processes and production scale of PHBV are still largely underdeveloped compared to the well-developed production of petrochemical polymers. It is predicted that further optimization of the PHBV production technology and extension of its production scale (eg to 50,000 tons per year) may result in an improvement in the condition of the environment. In addition, the results of the work show that the use of renewable instead of fossil electricity and heat resources required for PHBV production will ensure an effective optimization of the process. These results confirm the view that thanks to the expanding trends in the development of bioplastics production and the consumption of electricity and heat or changes in the sources of their generation, the bioplastics production industry can be significantly independent of fossil fuel products.

  1. Introduction: There is a lot of work available on both wood/biodegradable plastic composites as well as natural fibers/ biodegradable plastic composites in the literature (e.g. 10.1016/j.indcrop.2014.08.034, 10.1177/0892705718816354, 10.1515/secm-2016-0072). Also with PHBV matrix, even when the fillers are not identical, most bast fibers have similar compositions and properties, therefore it must be mentioned (e.g. 10.1007/s10924-009-0127-x, 10.1002/app.50182, 10.1016/j.compscitech.2012.01.021, 10.1007/s10924-020-01884-8, 10.1007/s10924-020-01979-2, 10.3389/fmats.2019.00275, 10.1016/S0266-3538(03)00102-7, 10.3390/su11082411, 10.1002/adv.21789, 10.3390/molecules24193538, 10.1016/B978-1-78242-373-7.00008-1, 10.1007/s10924-020-01781-0, 10.1016/j.compositesa.2019.05.009, 10.1002/9783527820078.ch8). It is mentioned in the discussion part briefly but not in the introduction.

Thank you for your attention, some of the indicated works have been added to the References and described in the article.

  1. Introduction: another very important disadvantage of these natural fillers is that their hydrophilic nature prevents them from having strong interfacial interactions that ultimately result in gaps and holes and premature mechanical failure. A lot of work has been conducted on functionalization and compatibilizers to improve their properties that must be at least mentioned (e.g. https://doi.org/10.3390/ma9040303 and work of the authors 10.3390/polym13121965, 10.12913/22998624/135399)

Thank you for your attention, some of the indicated works have been added to the literature and described in the paper.

  1. What is the difference of the PHBV hemp composites of this work with the previous work of the authors (10.12913/22998624/135399)

In the suggested study (10.12913 / 22998624/135399), the aim was to optimize the production process of the PHBV-hemp fiber biocomposite using Taguchi orthogonal plans in order to minimize processing shrinkage and maximize mechanical properties. The emphasis of the work focused on the assessment of the effectiveness and practical application of the indicated optimization method in the processing of the tested biocomposite.

On the other hand, in the reviewed work, the aim is to assess the influence of the type of filler on the processing properties, mechanical and functional, of biocomposites with the PHBV matrix. The fillers tested, i.e. wood, flax and hemp fibers, were selected due to their availability and popularity in the authors' geographical area of residence (Europe).

  1. Why was 30 wt% filler content chosen?

The filler content in the polymer matrix was selected as 30% due to the fact that in most cases the best mechanical, processing and functional properties were obtained from among the three tested contents (15%, 30%, 45%) - a small fragment of the results for the PHBV-hemp fiber composite is described in the paper : https://doi.org/10.1007/s10973-020-10492-6. In addition, it is expected that a comprehensive separate publication will be prepared showing the impact of the filler content (15%, 30%, 45%) on the mechanical, processing and functional properties of PHBV-fiber biocomposites: hemp, flax, wood.

  1. Section 2.1: What is the composition of the fibers used?

As mentioned in this chapter, 3 PHBV matrix composites were produced. Each contained a different type of filler in the amount of 30 wt.%. These were flax, hemp and wood fibers. The fiber supplier does not specify the specific fiber composition, but based on the literature data, the fiber composition is listed in the table below.

Tab. X. Mass fraction of individual components of plant fibers [Faruk O., Bledzki A.K., Fink H.P., Sain M .: Biocomposites reinforced with natural fibers: 2000–2010. Progress in polymer science, 37 (2012), 1552-1596; Kim J.K., Pal K .: Recent advances in the processing of wood-plastic composites. Springer Science & Business Media, Berlin 2010; Klyosov A.A .: Wood-plastic composites. John Wiley & Sons, New Jersey 2007.]

Type of fiber

Celluloze
(% mas.)

Hemicelluloze
(% mas.)

Lignin
(% mas.)

Others
(% mas.)

Bamboo

26–43

30

21–31

Linen/Flax

71

18,6–20,6

2,2

1,5

Kenaf

72

20,3

9

Jute

61–71

14–20

12–13

0,5

Hemp

68

15

10

0,8

Abaca

56–63

20–25

7–9

3

Sisal

65

12

9,9

2

Pineapple

81

12,7

Wheat straw

38–45

15–31

12–20

Rice straw

41–57

33

8–19

8–38

Deciduous trees

44 ± 3

32 ± 5

18 ± 4

0,2–0,8

Conifiers

42 ± 2

26 ± 3

29 ± 4

0,2–0,8

  1. Section 2.3: Natural fibers are very hydrophilic; where they dried before extrusion?

In order to minimize the presence of moisture / water in the obtained mixtures (PHBV-flax fibers, PHBV-hemp fibers, PHBV-wood fibers), they had to be dried before the extrusion process. Both hemp, flax and wood fibers as well as PHBV biopolymer tend to absorb water, which may have a negative impact on the possibility of obtaining a homogeneous structure of the composite - fiber fractions and PHBV with a high degree of water absorption may tend to clump into larger agglomerates. Moreover, drying should be performed in order to reduce the occurrence of possible air bubbles in the cross-section of the obtained pomace. Drying was carried out in a Chemland DZ-2BC laboratory dryer with a maximum power of 1400W and a capacity of 52l, equipped with a Value model V-i120SV pressure pump with a flow of 51l / min. The material to be extruded was dried at a temperature of 90 ° C for 6 hours with a vacuum in the drying chamber of 0.02 MPa.

  1. Section 3.1: Was there a significant effect of the fiber type on shrinkage? Where do these results stands in comparison with other data from the literature if available? Why did the presence of fibers reduce shrinking?

The use of fillers (not only of plant origin) in the polymer matrix results in a reduction of shrinkage of composites with a matrix of thermoplastic polymers, which, as we know, due to their properties, are characterized by processing shrinkage depending on their type (amorphous and partially crystalline materials). By adding a filler to the polymer matrix, we reduce the shrinkage of injection details. In the case of fibers, their geometry is characterized by a specific ratio of length to fiber diameter (L / d), which translates into a significant reduction in longitudinal shrinkage. During the flow of the plasticized polymer composite in the channels of the forming cavity, the fibers arrange along the flow direction - this results in the formation of a specific reinforcement of the polymer matrix counteracting shrinkage of the composite in the longitudinal direction [https://doi.org/10.1016/j.matdes.2012.04.058, https://doi.org/10.1016/j.dental.2013.04.016, https://doi.org/10.1016/j.matdes.2017.07.032, 10.12913 / 22998624.1120308] .

  1. Section 3.2: Was there a significant effect of the fiber type on water adsorption and why?

As can be seen in the figure on water absorption, the difference in water absorption between the composites is small. This is due to the similar morphology of plant fibers (Tab. X. Presented in the discussion). It should be noted, however, that there was a significant increase in water absorption of composites compared to pure PHBV, which is an obvious phenomenon.

Hydroxyl groups (OH) in cellulose, hemicellulose and lignin build a large amount of hydrogen bonds inside the macromolecule and between macromolecules in the cell wall of plant fibers. The action of water on plant fibers causes these bonds to break. The hydroxyl groups then form new hydrogen bonds

with water molecules that promote the swelling of the fiber. The swelling of the cell wall generates very high forces. The theoretical value of the pressure may be about 165 MPa [Stamman A.J .: Wood and Cellulose Science. Ronald Press, New York 1964.], however, the actual swelling pressure is half the calculated value [Tarkow H., Turner H.D .: The swelling pressure of wood. Forest Products Journal, 8 (1958), 193-197; Kondo T .: Hydrogen bonds in regioselectively substituted cellulose derivatives. Journal of Polymer Science Part B: Polymer Physics, 32 (1994), 1229-1236.]. Cellulose fibers interact with water not only on the surface but also in the entire volume.

The structure of cellulosic materials consists of crystalline and amorphous regions. Amorphous areas easily absorb chemical compounds, such as dyes and resins, and the presence of crystalline areas makes chemical penetration difficult [Hearle J.W., Morton W.E .: Physical properties of textile fibers. Woodhead Publishing, Manchester 2008.].

The possibility of water absorption by cellulose fibers depends, among others, on from:

  • Purity of cellulose: Raw cellulose material, such as unwashed sisal fibers, absorbs at least twice as much water as washed fibers due to the 24% pectin content.
  • Degree of crystallinity: all the OH groups in the amorphous phase are available to water, while only a small amount of water interacts with the surface OH groups of the crystalline phase.

The main disadvantage of cellulose fibers is their highly polar nature, which makes them incompatible with non-polar polymers. The poor resistance to water absorption makes the use of natural fibers less attractive for details that are to be used under external factors (e.g. rain, snow, hail). [Kalia, S .; Kaith, B.S .; Kaur, I. Cellulose Fibers: Bio- and Nano-Polymer Composites: Green Chemistry and Technology; Springer Science & Business Media: Berlin / Heidelberg, Germany, 2011 .; Kim, J.K .; Pal, K. Recent Advances in The Processing of Wood-Plastic Composites; Springer Science & Business Media: Berlin / Heidelberg, Germany, 2010 .; Smole, M.S .; Hribernik, S .; Kureˇciˇc, M .; Krajnc, A.U .; Kreže, T .; Kleinschek, K.S. Surface Properties of Non-Conventional Cellulose Fibers; Springer International Publishing: Cham, Switzerland, 2019.]

  1. Section 3.3: Was the difference in mechanical properties in the presence of the fillers statistically significant?

For each study of mechanical properties, a package of 7 test specimens was prepared for each type of composite in order to prepare the subsequent statistical treatment. For all test results of mechanical properties, the following were calculated: arithmetic mean, standard deviation and coefficient of variation. The results taking into account statistical data have been included both in tables and graphs (error bars). The differences in statistically significant properties are described in the chapters on the analysis of results and discussion.

  1. Section 4: Why was the properties in this work improved and what was the overall effect of the different filler types? The obtained results must be explained considering the chemistry of the materials as well as the data obtained from previous work reported in the literature.

Thank you for your attention. We tried to improve this in the analysis of the results, discussions and conclusions.

  1. The conclusions section must be shortened significantly to include only the most important information and provide insight as to why these results were obtained.

Thank you for your attention. We tried to improve it.

  1. Please remove the work “the” on the titles of the sections. Finally, I would like to thank the authors for considering my comments and wish them the best for their future research.
  2.  

Thank you for your attention. We tried to improve it.

WE HOPE THAT WE HAVE ENJOYED EVERY ATTENTION, CORRECTING THE MANUFACTURER AND ANSWERING QUESTIONS. ALL THE CHANGES MADE TO THE MANUSCRIP ARE MARKED WITH A RED FONT. THANK YOU FOR A DEEP ANALYSIS OF THE CONTENTS OF OUR MANUSCRIPT. ALL THE INDICATED COMMENTS WERE IMPORTANT AND ALLOWED TO INCREASE THE CONTENT VALUE OF OUR PUBLICATION. ALSO, WE WISH YOU THE BEST IN FUTURE RESEARCH, PUBLICATIONS AND REVIEWS.

Kind regards,
Wiesław Frącz 

Round 2

Reviewer 1 Report

I would like to thank the authors for their effort in improving the English level and content of the manuscript. My comments are as follows:

  1. There are still some English mistakes throughout the text. Line 22-24, the verb "were" must be changed to "was". In line 16, "biocomposite" must be changed to ""biocomposites" as they produced several biocomposites. Line 112-113 does not sound correct in relation with "The advantages of fibrous fillers include:". Line 69:  “Copolymerization with… leads to decrease the polymer degree of crystallinity” should be changed to “Copolymerization with… leads to decrease in/of the degree of crystallinity of the polymer". There are probably more mistakes so i suggest another control of the overall manuscript.
  2. Moreover, there are some punctuation mistakes such as in line 55, ".[5]."; line 186 ". [79-81].".
  3. I also suggest to remove the references at the end of the sentence in line 92, 161,164, 408, 426, 432. In line 408 "In the paper [80] the..." sounds very poor, i suggest the use of "Keller discussed the influence of...." and give the reference at the end of the sentence. The same thing is also valid for references 96, 97. So, another careful check is needed.
  4. The introduction is too long. 4 pages of introduction is too much for such a research article, it resembles to a review. The conclusion is indeed long. I would suggest to shorten them both, the table in introduction can be shorten also.

Author Response

Dear Sir/Madam,

    All your comments have been taken into account in the text. The changes introduced in the text are marked in purple. In some cases, the text has been removed at the request of one reviewer. The request of the second reviewer to correct the same fragment of the text could not therefore be taken into account. The article was also shortened and linguistically corrected. Thank you for your valuable comments. We hope you will be satisfied with our corrections.

Kind regards,

Wiesław Frącz

Reviewer 2 Report

The authors have significantly improved the content and overall quality of their manuscript. However, I respectfully believe that the potential of this work and especially its novelty and contribution in the field of biocomposites is not fully clear. While it was explained in depth in the replies to the comments and I do understand it, some of these explanations must be added in the manuscript. Another crucial issue is the size of the conclusions section; it is even bigger now. Below you will find more specific comments that should be addressed before publication.

Specific comments

  1. Abstract lines 14-19: The phrases used, in particular “the work is a part of the current policy...” and “this is especially true of the packaging industry” are not grammatically correct. Possible alternatives would be e.g. “This work is inspired by the current European policies that aim to reduce plastic waste”. The second sentence is unnecessary and can be removed from the abstract altogether.
  2. Abstract line 16: “The biocomposite” should be “the biocomposites”, since many composites were prepared.
  3. Abstract line 20-21: The word “technological” should be replaced by the word “technical”.
  4. Abstract lines 22-24: “A significant improvement of some mechanical properties of biocomposite…were obtained in comparison to pure PHBV” must be corrected to “A significant improvement of some mechanical properties of biocomposites … was obtained in comparison with pure PHBV”. Otherwise, the phrase should be “significant improvements…were obtained”.
  5. Introduction line 40: Perhaps the word “actions” has the meaning of “requirements”?
  6. Introduction lines 46-47: PHAs are not made from renewable raw materials that come from microorganisms; they are directly produced by microorganisms in the form of granules. This sentence must be revised accordingly.
  7. Introduction line 69: The phrase “Copolymerization with… leads to decrease the polymer degree of crystallinity” should be corrected to “Copolymerization with… leads to decrease of the pdegree of crystallinity of the polymer”.
  8. Introduction line 92: Delete “in”.
  9. Introduction line 140: Delete the first “the”.
  10. Introduction line 206: “Such as” can be replaced with “namely”, since those are all the 3 types of fibers used.
  11. The explanation provided for the novelty in the replies to my previous comments “The aim of the study was to assess the impact of the type of filler on the processing, mechanical and functional properties of biocomposites with the PHBV matrix - such comprehensive work on the above-mentioned fillers and matrix have not been found in the literature. The fillers tested, i.e. wood, flax and hemp fibers, were selected due to their availability and popularity in the authors' geographical area of residence (Europe). No work was found comparing these three types of composites in terms of production, processing and evaluation of mechanical, processing and functional properties. The results of the research may have a scientific as well as application aspect, allowing to indicate the possibility of manufacturing utility products in the injection molding process of the above-mentioned composites, especially in Europe, where these fillers are easily available and popular” is in my opinion very good and part of it should be added in the last paragraph of the introduction, as it is not as clear from the manuscript as it is in this particular reply. Also, in order to not have a huge introduction, the fiber types that were not part of this work could be removed from table 1.
  12. The word “the” must be removed from the tiles of the sections. For example, “the fiber investigation” could be changed to “Investigation of fiber structure” and as for the rest, simply remove “the”.
  13. The conclusions section is still too long – it is most common for it to be 1-2 paragraphs long. Conclusions should provide an overview of the most important findings, without going too much into detail, and also highlight the contribution of the work on the field of biocomposites.

Finally, I would like to congratulate the authors for their dedication on their work and for taking the time to reply to my comments.

Author Response

(The authors gave the same response as above.)
